# A triangular model of fractal growth with application to adsorptive spin-coating of polymers

**Kenneth Mulder**[1]*, **Sophia M. Lee**[2], **Wei Chen**[2]

**1** Department of Mathematics and Statistics, Mount Holyoke College, South Hadley, Massachusetts, United States of America, **2** Department of Chemistry, Mount Holyoke College, South Hadley, Massachusetts, United States of America

* kmulder@mtholyoke.edu

**Editor:** Malgorzata J. Krawczyk, AGH University of Science and Technology Faculty of Physics and Applied Computer Science: Akademia Gorniczo-Hutnicza im Stanislawa Staszica w Krakowie Wydzial Fizyki i Informatyki Stosowanej, POLAND

## Abstract

Over the last 40 years, applied mathematicians and physicists have proposed a number of mathematical models that produce structures exhibiting a fractal dimension. This work has coincided with the discovery that objects with fractal dimension are relatively common in the natural and human-produced worlds. One particularly successful model of fractal growth is the diffusion limited aggregation (DLA) model, a model as notable for its simplicity as for its complex and varied behavior. It has been modified and used to simulate fractal growth processes in numerous experimental and empirical contexts. In this work, we present an alternative fractal growth model that is based on a growing mass that bonds to particles in a surrounding medium and then exerts a force on them in an iterative process of growth and contraction. The resulting structure is a spreading triangular network rather than an aggregate of spheres, and the model is conceptually straightforward. To the best of our knowledge, this model is unique and differs in its dynamics and behavior from the DLA model and related particle aggregation models. We explore the behavior of the model, demonstrate the range of model output, and show that model output can have a variable fractal dimension between 1.5 and 1.83 that depends on model parameters. We also apply the model to simulating the development of polymer thin films prepared using spin-coating which also exhibit variable fractal dimensions. We demonstrate how the model can be adjusted to different dewetting conditions as well as how it can be used to simulate the modification of the polymer morphology under solvent annealing.

## Introduction

In nature as well as in human-determined realms, a surprising number of structures exhibit self-similar and scale-invariant features. Historically, such structures sparked the interest of mathematicians as they defied our conventional notion of dimension. Dubbed "fractals" by Benoit Mandelbrot in his seminal book <u>The Fractal Geometry of Nature</u> [1], rather than playing by the normal rules of Euclidean geometry in which idealized objects (e.g. lines, planes,

**Data Availability Statement:** All relevant data are within the manuscript and its Supporting Information files.

**Funding:** SL was funded by the National Science Foundation, grant number DMR-1807186 (nsf. gov). The funders had no role in study design, data collection and analysis, decision to publish, or preparation of the manuscript.

**Competing interests:** The authors have declared that no competing interests exist.

and spheres) have integer dimension, these structures exhibit a fractional dimension. (In this work, fractal dimension will generally refer to Minkowski-Bouligan (or box-counting) dimension.) Since the emergence of the concept, fractal structures have been discovered in nearly every field of science [1,2]. Many physical materials exhibit a fractal structure [3–5], and bacteria and other organisms follow fractal growth patterns [6,7] as do lightning [8] and river networks [9]. Fractal patterns have been discovered on Mars [10] and in patterns of urban migration and development [11]. Dendritic growth is a particularly common form of fractal growth [2,12,13].

Due to the ubiquitous nature of fractal structures, including in the material sciences, mathematical models that produce fractal growth are of particular interest with many models focused on the aggregation of particles [2,14]. Early models include the Eden model of random growth on the boundary of a mass [15] and the Ballistic Aggregation model in which "sticky" particles are stochastically shot or dropped onto a surface [16]. Other, slightly more complicated models include the Dielectric Breakdown (DB) model [8] and the Born model of fracture growth [17]. (See [14] for an historical overview.) But arguably the most successful model of fractal growth has been the Diffusion Limited Aggregation (DLA) model of Witten and Sander [14,18].

The DLA model is notable for its stark simplicity, motivated by the physical dynamics of "sticky" particles diffusing in a medium [19]. The model begins with a centrally located particle followed by a sequence of introduced particles on random walks (Brownian motion), either on a lattice or in Euclidean space. Each particle keeps moving until it encounters a particle already incorporated into the aggregation. At this point, it stops moving and becomes part of the aggregate. Despite its simplicity, the DLA model has been successful in replicating numerous observed phenomena including Martian araneiforms [10], breast lesions [20], and thin film "islands" of one material grown on another [4].

DLA structures tend to exhibit dendritic growth and have a fractal dimension. The exhibited dimension when grown in two dimensional space is 1.71 and seems to be a constant even though the model exhibits a high degree of variability over a range of parameter values or through relaxation of some assumptions such as lowering the probability of sticking below unity [19,21]. Other, more extreme, modifications of the model can result in structures with different fractal dimensions. Modifying the nature of the random walk [22] or the probability structure of where particles enter the space [20] can produce significant variability in structure and dimension. A further modification is to allow the clusters themselves to diffuse, a so-called cluster aggregation model, which results in a significant reduction in dimension [23].

Other modifications to the DLA model have been problem specific. Wang et al. [9] modified the movement of the particles to reflect geographical constraints in order to allow the DLA model to predict river network formation. Xia [24] added specific orientation criteria to encourage production of needle-like structures. And the dielectric breakdown model can be replicated by adding an electric field to the DLA model thereby increasing the probability of attaching to an extending tip [8]. In all three cases, the general fractal structure of the DLA model is preserved while the output is modified to meet specific criteria. More importantly, the overall conceptual simplicity of the DLA model is largely preserved even if the computational complexity is increased.

Other models of fractal growth tend to be significantly more complex as they seek to incorporate continuous processes and capture a more nuanced picture of reality. Examples include models of frost formation that seek to include fluid flow and phase field dynamics (e.g. [25]) and models of bacterial growth that incorporate nutrient gradients [6]. Polimeno, Kim, and Blanchette [26] added several new factors to the DLA model with the goal of making it more consistent with observed empirical phenomena. Such models are important and often include

DLA-like mechanisms, but their greater realism comes at a cost of higher complexity and consequent reduction in tractability. As noted by many authors [27,28], we often learn the most from early, simple models, and gains in understanding and management can decline rapidly with increased complexity.

In this work, we seek to make an addition to the library of fractal growth models that is of intermediate complexity. It is more complex than the DLA model, but its definition and dynamics are conceptually and mechanistically straightforward. Here we introduce a 2D model of fractal growth that we believe is novel and different from cellular or particle-based models of growth and aggregation. Instead of modeling particles, we model the bonds formed by a growing mass as it incorporates particles from the surrounding medium. Geometrically, a particle attaches to the mass by linking to the two endpoints of an existing link forming a triangle. Additionally, as the mass is forming it is also contracting. Particles are initially stationary. When they bond to the mass, they become mobile and are pulled toward any particles they are attached to. This contraction process continues until the mass solidifies and further movement ceases.

Below we introduce the model, explore its properties including a variable fractal dimension, and give an example of its application by showing how it replicates the growth of dewetted polymer structures.

## Model definition

### Model structure

Our model consists of two primary components—points in the plane (representing particles) and triangles consisting of three linked particles which represent bonds connecting these particles. The model is created iteratively by attaching a previously unattached particle to both ends of an existing link thereby creating a new triangular bond. Links that are candidates for forming new triangular bonds are considered "active". When a new triangle forms, the initiating link is turned off while the other two links that form the triangle are now active.

**Model initiation.** Because our model frequently exhibits radial growth, and one of our applications is to adsorptive spin-coating of polymer solution, we begin with a random, uniform distribution of particles on a disk. The number of particles is set by a parameter, and the radius of the disk is set so as to achieve a specified particle density. Throughout this paper, unless otherwise mentioned, the model has 40,000 particles with an average density of 100 particles per square unit. This yields a radius of 11.3 units. One particle is placed at the origin and a starting triangle comprised of three links is formed by connecting that particle to its two closest neighbors. The three initial links are all set to be active.

The model runs iteratively, alternating between a growth step in which active links have the potential to bond to new particles and a contraction step in which attached particles adjust their position seeking to minimize the distance to their linked neighbors. The two steps continue to alternate until either all particles have been attached or no active links remain.

**Growth step.** Each time step, every active link has a specified probability (*bond-rate*) of seeking to bond with a previously unattached particle. This parameter determines the rate at which new bonds are formed under ideal bonding conditions. The number of links seeking to bond is determined by a random draw from a binomial distribution and the specified number of links is randomly chosen from all active links. Links seek to bond in a random order. When a link seeks to form a bond, the model searches for the unattached particle with the shortest average distance to the endpoints of the link. If the average distance to this particle is below a given threshold (*bond-range*), then both endpoints of the link form links with the new particle forming a new triangular bond. Each new link is then activated. Regardless of whether a bond

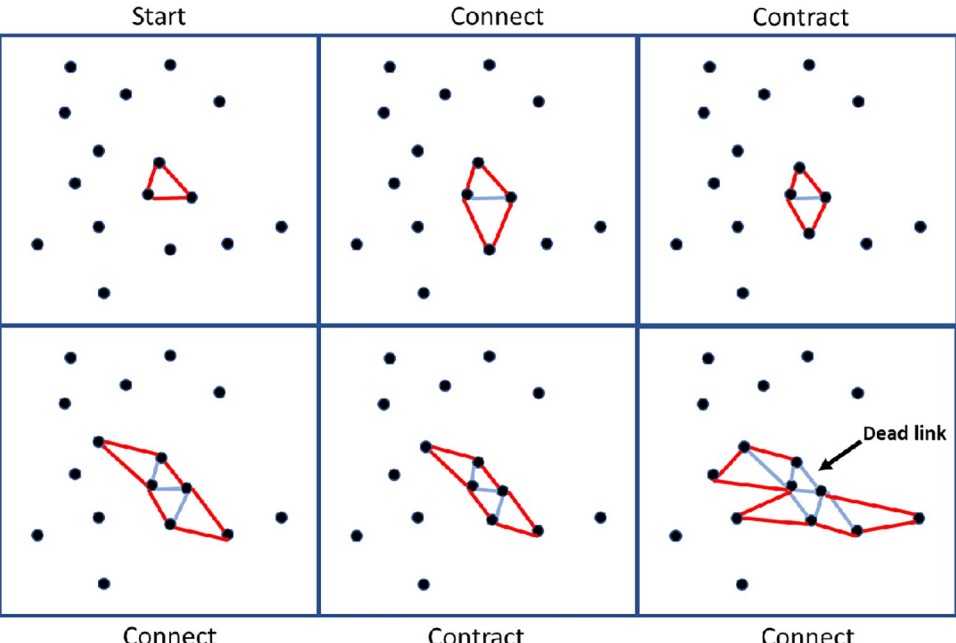

**Fig 1. First 3 growth steps and first 2 contraction steps for a stylized example model.** Each connection step, active links are randomly selected (colored to blue) and connected to the nearest particle if the particle is within bonding range. The "dead link" in the last window attempted to bond but had no unattached particles within the bonding range. In the contraction step, all attached particles that are mobile experience a force in the direction of the particles they are attached to which leads to the network contracting. The amount of contraction has been exaggerated for demonstration purposes.

is formed, the initiating link is deactivated and cannot be activated again. If a link is deactivated without bonding, we refer to it as a dead link. Otherwise, it becomes an internal link.

**Contraction step.** After the growth step is complete and all selected links have been deactivated, the model experiences a contraction step as if the links between particles were ideal springs. When a particle is connected to the network, it is mobile for a specified number of time steps (*mobility*) after which it remains fixed in location. During the contraction phase, all mobile particles are pulled closer to their link neighbors (particles they are linked to) by experiencing a displacement given by $\Delta\vec{r} = 0.01 \times \sum_{i\in N}\vec{r}_i$ where $N$ is the set of all particles linked to the target particle and $\vec{r}_i$ is the vector extending from the target particle to particle $i$. We consider this to be a discrete approximation to a continuous contraction process.

When an attached particle has experienced contraction for a number of time steps equal to the mobility, it then becomes fixed in its location though it will continue to exert a force on any of its link neighbors that are still mobile. A stylized representation of the initial growth and contraction steps is shown in Fig 1. When no new triangular bonds are possible because no unattached particles remain within the bonding range, the model then continues to experience contraction for a number of time steps equal to the mobility thereby allowing recently attached particles to experience full contraction. Model parameters are reported in Table 1.

## Example models

Figs 2 and 3 show a series of intermediate stages and the final output for two different runs of the model. (All model images reported in this paper will show the links connecting particles since it is the bonds that are being modeled. The particles themselves are hidden.) For demonstration purposes, these models are smaller (5,000 particles) than the other models shown in

**Table 1. Model parameters and their definitions.**

| Parameter | Abbreviation | Definition |
|---|---|---|
| Number of particles | *n-part* | Number of randomly arranged particles in the original disk. |
| Particle density | *density* | Average density of particles per square unit. |
| Probability of bonding | *bond-rate* | Probability that an active link seeks to bond to a new particle each time step. |
| Maximum bonding distance | *bond-range* | Maximum average distance to the endpoints of a link for a particle to be able to bond to the link. |
| Mobility period | *mobility* | Number of time steps after bonding during which a particle experiences contraction. |

the paper (40,000 particles) allowing the entire growth process to be seen. The masses generally grow radially, although some branches in Fig 2 extend back in toward the center to reach unattached particles. As the model grows, branches closer to the center complete their contraction while branches at the outer edge display the denser structure that precedes full contraction. The maximum bonding distance is greater in Fig 3 which results in fewer dead links and more direct, radial growth.

## Model behavior

**Determinants of behavior.** Model behavior is primarily dictated by two factors. The first is the average growth rate in active links as a function of the number of attached particles. This is directly related to the probability that an active link forms a new bond when it is selected. In order for a link to bond, an unattached particle must be within the maximum bonding distance (*bond-range*). This is in part determined by the ratio of *bond-range* to the average distance between particles at the start of the model. In this analysis, the latter is kept constant by keeping the particle density constant, thus *bond-range* is the sole parameter governing this

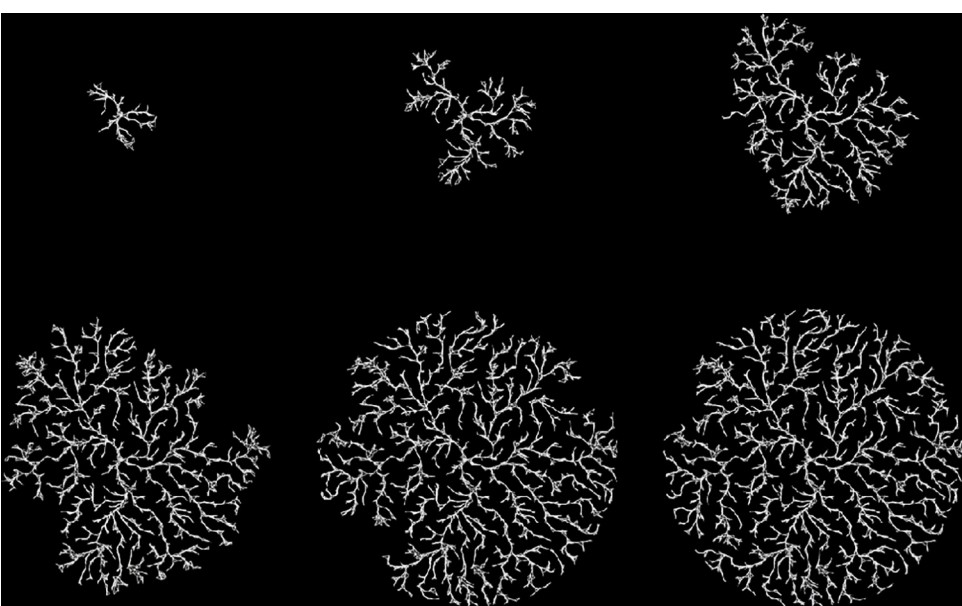

**Fig 2. Example growth in a model with 5000 particles.** Parameter values are *bond-rate* = 0.05, *mobility* = 50 and *bond-range* = 0.2. Output is shown at times 150, 300, 450, 600 and 963 (the final time). The final image (lower right) is the output at time 963 zoomed in to demonstrate the triangular nature of the structure.

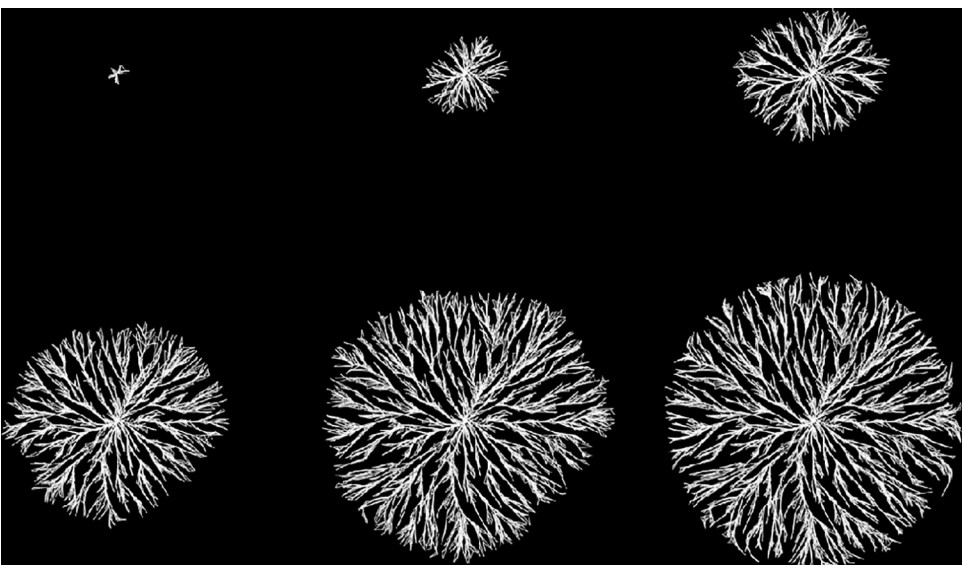

**Fig 3. Example growth in a model with 5000 particles.** Parameter values are *bond-rate* = 0.02, *mobility* = 50 and *bond-range* = 0.5. Output is shown at times 150, 300, 450, 600 and 766 (the final time). The final image (lower right) is the output at time 766 zoomed in to demonstrate the triangular nature of the structure.

ratio. However, each newly attached particle produces two active links, an exponential growth process that outpaces the growth in the number of particles within a given range as the structure grows. Thus, as closer unattached particles become linked, the required bonding distance steadily increases and will reach a point at which some links fail to bond and become dead links.

Contraction of the model also causes the distance between active links and unattached particles to increase as mobile particles and their links are pulled toward the center. This will increase the probability of a dead link forming. In particular, how many contractions the endpoints of a link experience before that link seeks to bond determines how far toward the center and away from unattached particles a link moves. The number of contractions a link experiences before attempting to bond is determined by how long it waits to be selected for bonding (inversely proportion to *bond-rate*) and the *mobility* (which determines the total number of contractions a particle experiences). If *mobility* is low, then active links do not move very far regardless of the *bond-rate*. But if *mobility* is high and the *bond-rate* is low, then links may experience significant contraction before seeking to bond potentially moving beyond the *bond-range*.

The second determinant of model behavior is the amount of contraction that takes place after bonding which is determined by *mobility*. Models with high *mobility* experience significant contraction which leads to shorter side branches and lower density in the final structure. Models with low *mobility* have longer side branches and higher density.

Together these two determinants lead to a heuristic for predicting model growth. First, the degree of curliness and spatial variability caused by dead links can be adjusted by increasing or decreasing the active link growth rate. This can generally be effected by adjusting the *bond-range* although changing the *bond-rate* can also affect this if the *mobility* is sufficiently high. Once the proper growth rate has been determined, the relative density and average length of branches in the structure can be adjusted by varying the *mobility*.

**Descriptive model statistics.** We calculate five statistics that describe model growth and final structure. First, we directly estimate the active link growth rate. If there are no limits to

bonding, then every active link bonds and each newly bonded particle adds two active links. However, generally one or more links fail for each link that succeeds in attaching to a new particle. We calculate the ratio of the change in active links divided by the change in attached particles between 5000 and 10,000 timesteps. Generally, we see an average increase in active links per attached particle of between 0.025 and 0.2. However, if too many links are unable to bond, the growth rate can be negative and the structure will fail to expand into all areas and some particles may fail to attach. Consequently, as a second statistic, we calculate the percentage of initial particles that become attached, referred to as percent attached.

Third, since unattached particles lie outside the area of current growth, when the growth rate in active links is high, we expect the growth to have a strong, radial pattern. Conversely, when more deadlinks appear, we expect growth to show greater variability in direction and thus be less radial. One way to measure radial growth is to estimate the growth rate in the number of links intersecting a concentric circle of the disk as a function of the radius of the circle. This will generally be a linear function of the radius, but the slope of this line will be lower for networks that are curvier with less direct radial motion. Plotting the number of intersecting branches as a function of radius for a variety of models supports this claim. We refer to this statistic as the branch-slope.

Fourth, we calculate the variability in network structure by calculating the standard deviation in the number of links each particle is connected to, referred to as the standard deviation of the network degree. Finally, to assess the overall level of contraction, we report the average branch length. Higher mobility should lead to shorter branch lengths. However, this should also be influenced by *bond-range* with greater bonding ranges leading to longer links.

**Example models.** This leads to two primary axes describing model behavior. The first axis is *bond-range*, which dictates how direct the radial growth is. Low radial growth results in curvier branches and a higher proportion of dead links. The second axis is *mobility*, which controls the relative density and how long the side branches are. Fig 4 shows nine examples of model output arranged according to these two axes. As the maximum bond range increases, growth becomes straighter and more radial. As the mobility increases, the side branches become shorter and the overall structure becomes less dense.

Fig 5 shows heat maps of five model statistics and fractal dimension for five values of *mobility* and seven values of *bond-range*. *Bond-rate* was set to 0.1. The maximum bonding range increases from left to right and the mobility increases from top to bottom in each map. Parameter values were chosen to give provide a thorough coverage of parameter phase space based on exploratory analysis.

All five model statistics are significantly impacted by *bond-range*. Only branch-slope and active branch length are affected by *mobility*. The growth rate, the proportion of particles attached, and the variability in the network degree of each particle all increase with *bond-range* and are unaffected by mobility. The branch-slope increases with *bond-range* as growth becomes more radial and diminishes with *mobility* as the side branches become shorter. The same is true for average branch length. Similar figures were calculated for bonding rates of 0.02 and 0.5 (not shown). The same patterns with regard to *bond-range* and *mobility* were observed. The growth rate, branch-slope, and average branch length all increased with the bonding rate. The proportion of particles attached increased when *bond-rate* went from 0.02 to 0.1 but did not change when it was increased further to 0.5. The variability in network degree increased with *bond-rate* for lower levels of *bond-range* and decreased at higher levels of *bond-range*.

When the growth rate in active links is too low, the model has the potential to die out before utilizing all (or even most) of the particles, and even when the space is fully colonized, the path taken from the center to a branch tip can be very circuitous. Fig 6 shows two such outcomes,

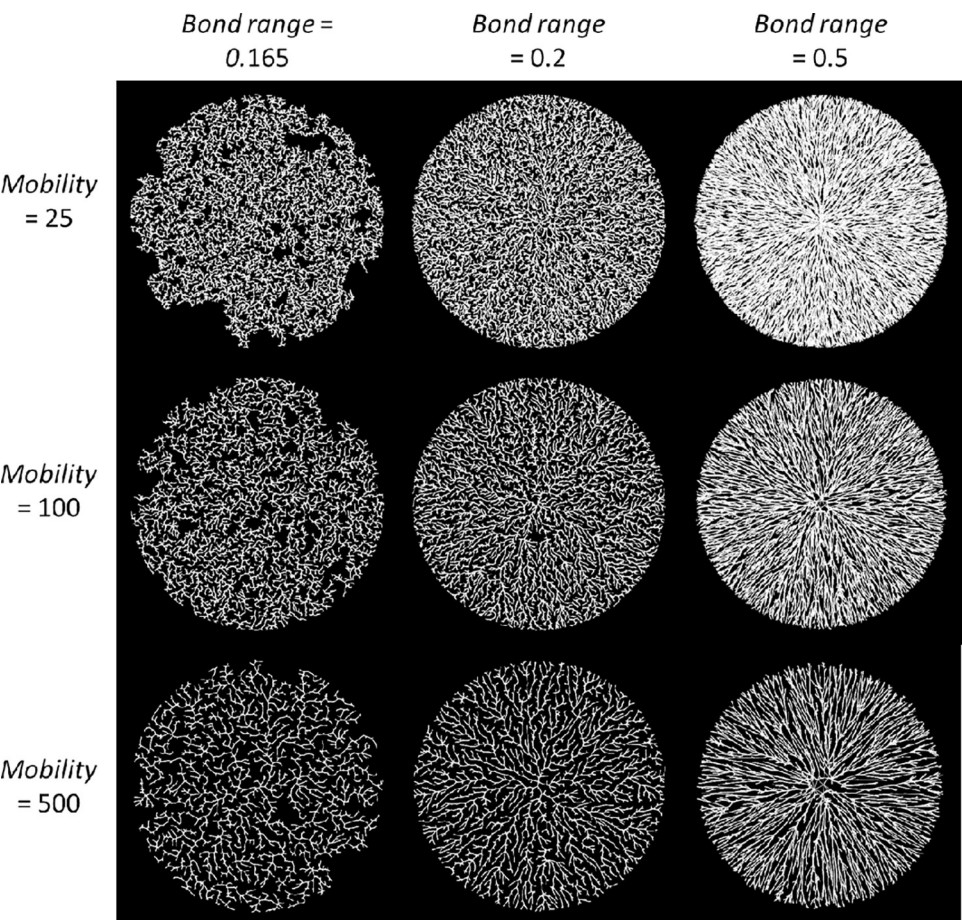

**Fig 4. Nine representative final structures for our model arranged along two axes.** The vertical axis controls the mobility which dictates the degree of contraction the structure experiences as it grows. The horizontal axis controls the maximum bonding range which dictates the growth rate in active links and subsequent radial growth.

one with a lower *mobility* (higher density) and one with a higher *mobility* (lower density). In both situations, the average growth rate is barely positive leading to a high proportion of dead links. As a result, growth is often more similar to a random-walk than it is to radial growth. Indeed, in the model with higher *mobility*, initial growth was restricted to just two branches and as a result the network is comprised of two distinct clusters. The model with lower *mobility* is composed of only three clusters although this is more difficult to discern visually.

## Fractal nature of model growth and variability in the fractal dimension

Image analysis using the box-counting method for estimating fractal dimension [29] shows model output to consistently have a fractional dimension. Because model output consists of triangular bonds, dimension could not be estimated using the coordinates of the particles (the vertices of the bonds). Instead, model output was converted to a binary image and dimension was estimated using the Fraclac plugin [30] in the image processing software ImageJ [31]. To ensure dimension was comparable across models, image scale and resolution were kept constant.

Analysis across models shows the mean and variance of the fractal dimension both vary with parameter settings, a feature somewhat unusual for a relatively simple growth model.

**Active Link Growth Rate**

| | | | | | | |
|---|---|---|---|---|---|---|
| 0.031 | 0.042 | 0.045 | 0.053 | 0.080 | 0.100 | 0.176 |
| 0.027 | 0.040 | 0.044 | 0.053 | 0.078 | 0.099 | 0.178 |
| 0.024 | 0.043 | 0.047 | 0.053 | 0.080 | 0.097 | 0.180 |
| 0.025 | 0.042 | 0.049 | 0.052 | 0.080 | 0.100 | 0.174 |
| 0.028 | 0.035 | 0.047 | 0.051 | 0.078 | 0.101 | 0.175 |

**Average Branch Length**

| | | | | | | |
|---|---|---|---|---|---|---|
| 0.085 | 0.089 | 0.093 | 0.100 | 0.120 | 0.141 | 0.226 |
| 0.071 | 0.074 | 0.078 | 0.082 | 0.098 | 0.114 | 0.181 |
| 0.060 | 0.063 | 0.065 | 0.069 | 0.081 | 0.094 | 0.146 |
| 0.047 | 0.049 | 0.050 | 0.053 | 0.062 | 0.070 | 0.105 |
| 0.042 | 0.043 | 0.045 | 0.047 | 0.055 | 0.062 | 0.090 |

**Branch-Slope**

| | | | | | | |
|---|---|---|---|---|---|---|
| 62.174 | 72.560 | 79.327 | 90.260 | 123.380 | 153.960 | 268.940 |
| 50.186 | 60.340 | 67.860 | 75.560 | 102.880 | 127.320 | 219.280 |
| 42.511 | 51.500 | 57.460 | 64.180 | 87.180 | 104.700 | 181.320 |
| 33.457 | 40.531 | 46.245 | 53.480 | 68.680 | 82.340 | 136.840 |
| 29.652 | 36.298 | 41.940 | 48.060 | 62.420 | 73.620 | 120.360 |

**St. Dev. of Network Degree**

| | | | | | | |
|---|---|---|---|---|---|---|
| 1.681 | 1.708 | 1.730 | 1.754 | 1.798 | 1.826 | 1.874 |
| 1.690 | 1.718 | 1.737 | 1.759 | 1.802 | 1.829 | 1.876 |
| 1.689 | 1.717 | 1.739 | 1.760 | 1.802 | 1.829 | 1.875 |
| 1.689 | 1.718 | 1.737 | 1.759 | 1.802 | 1.830 | 1.875 |
| 1.690 | 1.717 | 1.737 | 1.759 | 1.802 | 1.828 | 1.876 |

**Percent Attached**

| | | | | | | |
|---|---|---|---|---|---|---|
| 0.877 | 0.970 | 0.990 | 0.998 | 1.000 | 1.000 | 1.000 |
| 0.862 | 0.966 | 0.990 | 0.998 | 1.000 | 1.000 | 1.000 |
| 0.857 | 0.969 | 0.989 | 0.998 | 1.000 | 1.000 | 1.000 |
| 0.861 | 0.968 | 0.990 | 0.998 | 1.000 | 1.000 | 1.000 |
| 0.852 | 0.968 | 0.990 | 0.998 | 1.000 | 1.000 | 1.000 |

**Fractal Dimension**

| | | | | | | |
|---|---|---|---|---|---|---|
| 1.771 | 1.788 | 1.791 | 1.792 | 1.793 | 1.793 | 1.793 |
| 1.787 | 1.783 | 1.787 | 1.787 | 1.791 | 1.791 | 1.790 |
| 1.754 | 1.777 | 1.781 | 1.783 | 1.786 | 1.786 | 1.788 |
| 1.733 | 1.758 | 1.763 | 1.766 | 1.772 | 1.775 | 1.778 |
| 1.721 | 1.746 | 1.752 | 1.756 | 1.762 | 1.765 | 1.773 |

**Fig 5. Heat maps and values for five model statistics and the fractal dimension of the final structure.** *Bond-rate* = 0.1. From left to right in each map, *bond-range* values are 0.165, 0.175, 0.185, 0.2, 0.25, 0.3 and 0.5. From top to bottom in each map, *mobility* values are 25, 50, 100, 300, and 500. Heat maps are color coded according to the range of each variable with the lowest values shaded green and the highest values shaded red. The values are the averages for 50 model runs for each combination of parameters. Data are available in S1 File.

Observed fractal dimensions range from 1.50 to 1.84 with most structures having a dimension greater than 1.70. Average fractal dimensions for seven different bonding ranges and five different mobilities are shown in Fig 5. Fractal dimension increases with *bond-range* and decreases with *mobility*. At lower levels of *bond-range*, dimension increases with *bond-rate*.

Lower dimensions occur when the average growth rate is close to zero, which can be achieved generally by decreasing *bond-range* (Fig 5). When the maximum bonding range is greater than 0.2, fractal dimension is generally greater than 1.75. Lower fractal dimensions are

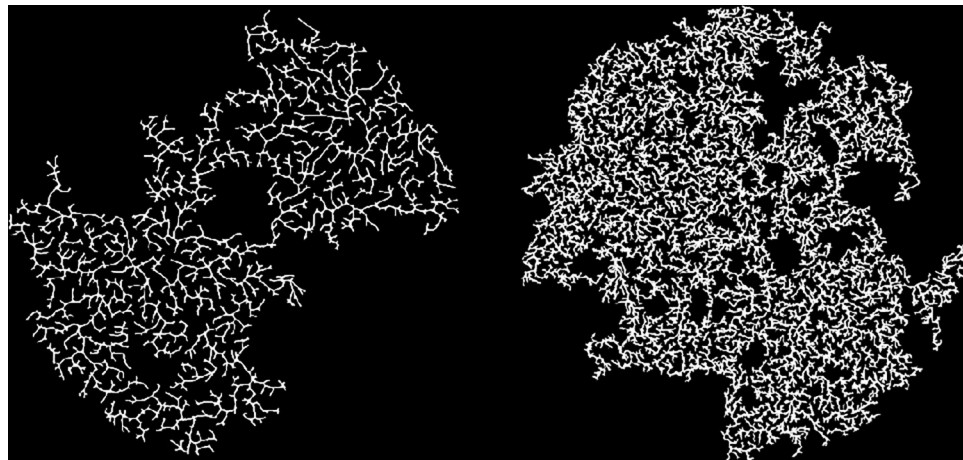

**Fig 6. Sample model output.** The average growth rate is ~ 0.015 which is near the lower end for viable growth or, on average, a slight increase in the number of active lengths for each new particle attached. Higher *mobility* (300) in the model on the left leads to a lower-density network while lower *mobility* (25) in the model on the right leads to a higher-density network. Parameter values for both runs are *bond-rate* = 0.1 and *bond-range* = 0.16.

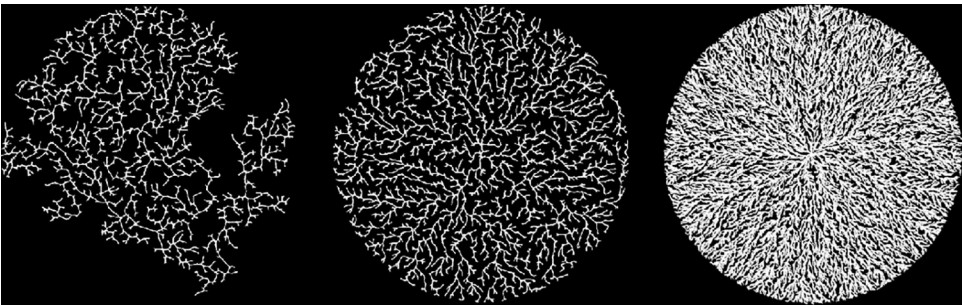

**Fig 7. Representative model output with three different fractal dimensions.** From left to right, the fractal dimensions using the box-counting method are 1.65, 1.75, and 1.79. Parameter values from left to right are as follows: {*bond-rate* = 0.64, *mobility* = 456 and *bond-range* = 0.15}, {*bond-rate* = 0.01, *mobility* = 31 and *bond-range* = 0.25}, and {*bond-rate* = 0.02, *mobility* = 19 and *bond-range* = 0.3}.

generally characterized by a high proportion of unattached particles. Fig 7 shows three representative model outputs with different fractal dimensions.

It is of some interest that the same set of model parameters can lead to significant variation in dimension. Fig 8 shows the range of fractal dimensions achieved over multiple runs when all parameters are kept fixed. For the runs shown, only *bond-rate* was varied, and multiple runs were conducted for each setting of the bonding rate. The average fractal dimension increases with the bonding rate, but significant variability is seen for low and moderate bonding rates where a high proportion of particles remained unattached. This is likely due to the higher variability of the negative binomial distribution with lower *bond-rate*. Links that wait longer to attempt bonding experience more contractions and thus a higher likelihood of failing to bond. At higher bonding rates, the variability in dimension decreases significantly.

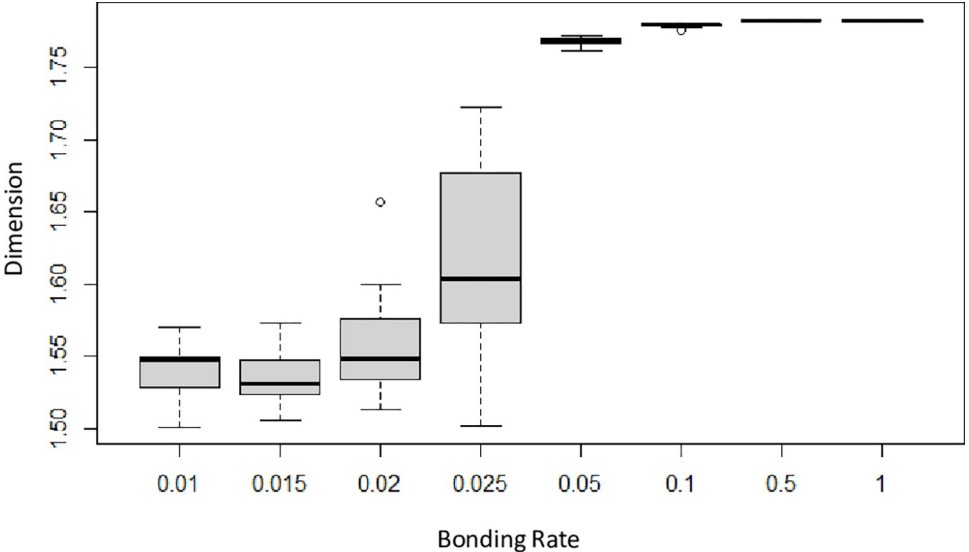

**Fig 8. Distribution of fractal dimensions for different values of bonding rate.** Other parameter settings are as follows: *bond-range* = 0.18, *mobility* = 100. The sample size varies as some runs were discarded due to a high proportion of unattached particles. Sample size was 7–13 for bonding rates less than 0.025 and 20 for the other groups. Data are available in S2 File.

## Simulation of polymer development

### Fractal polymer growth

Model output for our triangular growth model produces patterns very similar to those observed in the morphologies of polymer thin films prepared by adsorptive spin-coating. In particular, Qi et al. [32], by adjusting the amount of poly(vinyl alcohol) (PVOH[99%H]: 89–98 kDa and 99% hydrolyzed) solution deposited on a polydimethylsiloxane substrate (PDMS[49k]: 49 kDa), produced polymer patterns very similar to our model output both in terms of structure and fractal dimension (see Fig 9). Poly(vinyl alcohol) is synthesized by hydrolyzing poly(vinyl acetate) to convert a large portion of acetate groups to alcohol groups, -OH. On average ~99% of the repeat units in PVOH[99%H] consists of -OH groups, which are capable of hydrogen bonding interactions and crystallization. The remaining ~1% of the repeat units contains acetate groups, which can engage in hydrophobic interactions. The formation of the fractal features in the PVOH[99%H] thin films is driven by crystallization during the drying process [33]. PDMS[49k] is a hydrophobic, flexible support that facilitates the dewetting of the PVOH[99%H] thin films [32–34]. In the study by Qi et al., the PVOH[99%H] polymer thickness (amount of polymer per unit area) was the same under various experimental conditions, similar to the constant particle density in our model.

In Fig 9, there is a clear relationship between drop size and fractal dimension as well as with density. The primary cause of this relationship is the fact that smaller drops have a much longer drying time due to a greater proportion of the mass experiencing weaker centrifugal force. This gives particles more time to experience contracting forces prior to solidification. In the model, increased drying time is simulated by increased mobility, and we do indeed see a similar relationship between mobility and dimension/density (see Figs 4 and 5).

Moreover, Qi et al. observed an inverse relationship between the fractal dimension and the lacunarity of the images. Lacunarity ($\lambda$) was introduced by Mandelbrot [1] as a complementary measure to dimension for describing fractals. It captures the texture of a fractal by measuring the spatial variability in image density. To better compare our model output to the spin-coated polymers, we estimated lacunarity for a range of parameters using the Fraclac plugin [30] in ImageJ [31]. Our model also produces an inverse relationship between dimension and lacunarity that closely matches values reported by Qi et al. (Fig 10). Generally, we see a negative linear correlation between lacunarity and dimension. Model parameters varied across their full range with the exception of *bond-range*. When the bonding range was 0.2 or greater, output tended to cluster around a dimension of 1.77 to 1.79 and lacunarity between 0.28 and 0.3, values beyond the range of those observed in the polymer data in Fig 9. For the parameters chosen,

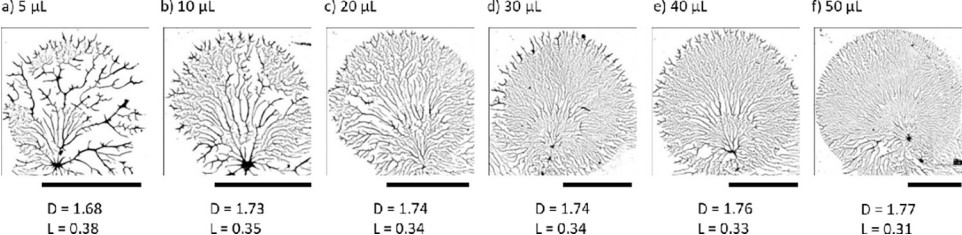

| a) 5 μL | b) 10 μL | c) 20 μL | d) 30 μL | e) 40 μL | f) 50 μL |
|---|---|---|---|---|---|
| D = 1.68 L = 0.38 | D = 1.73 L = 0.35 | D = 1.74 L = 0.34 | D = 1.74 L = 0.34 | D = 1.76 L = 0.33 | D = 1.77 L = 0.31 |

**Fig 9. Binary optical images of dewetted patterns obtained from adsorptive spin-coating.** 5–50 μL PVOH[99%H] solutions spin-coated on PDMS[49k] substrates (1.4 cm x 1.4 cm) at 6000 rpm. Each image (12.5x) was converted to binary after background subtraction so that fractal dimension (D) and lacunarity (L) values could be determined. The scale bars are 2 mm in length. Reprinted from Qi et al. [32] under a CC BY license, with permission from the American Chemical Society, original copyright 2019.

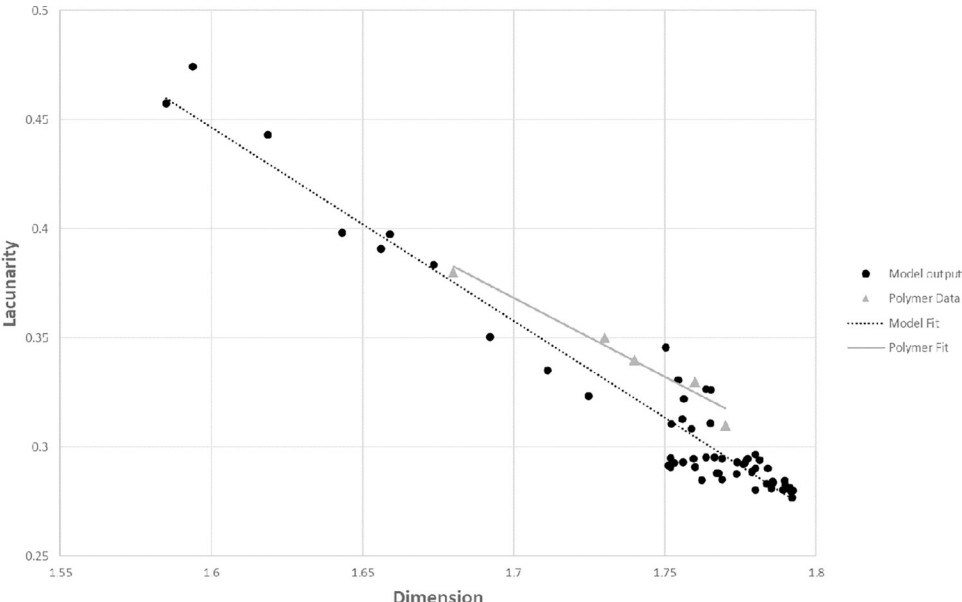

**Fig 10. Scatter plot of lacunarity versus dimension for model output and for six spin-coated polymers.** Model parameters were randomly drawn for each run. *Bond-rate* varied between 0.01 and 1.0, *mobility* between 15 and 500, and *bond-range* between 0.16 and 0.20. Two of the empirical samples had identical values. Data are available in S3 File.

while the sample size from Qi et al. is small, all six samples lie within the general range of values seen in Fig 10.

## Additional aspects of polymer growth

There are additional aspects of polymer pattern development that the model is able to capture. Fig 11 demonstrates two distinct annular regions in polymer pattern formation that result from a difference in spin rates and consequent drying times. The inner region experiences a longer drying time due to reduced centrifugal force leading to a circular boundary between a looser, lower dimensional inner core and an outer band displaying a tighter, higher dimensional pattern that results from a shorter drying time. Faster spin rates result in the region with shorter drying times extending farther in toward the center. Fig 12 shows a similar pattern produced by our model. The change is the result of a difference in mobility between particles that are above and below a fixed distance from the center. In particular, the radius of the figure is 11.3 units, and particles closer than 6.5 units to the center have a higher mobility than those

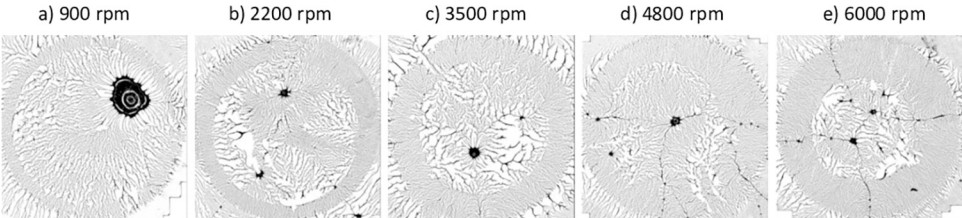

**Fig 11. Binary optical images of dewetted patterns after adsorptive spin-coating.** A 200 μL PVOH$^{99\%H}$ solution (drop diameter: 9.2 ± 0.1 mm) spin-coated on PDMS$^{49k}$ substrates (1.4 cm x 1.4 cm) at 900–6000 rpm. Each image was stitched together from sectional optical images (50x) using Autopano Giga. Reprinted from Qi et al. [32] under a CC BY license, with permission from the American Chemical Society, original copyright 2019.

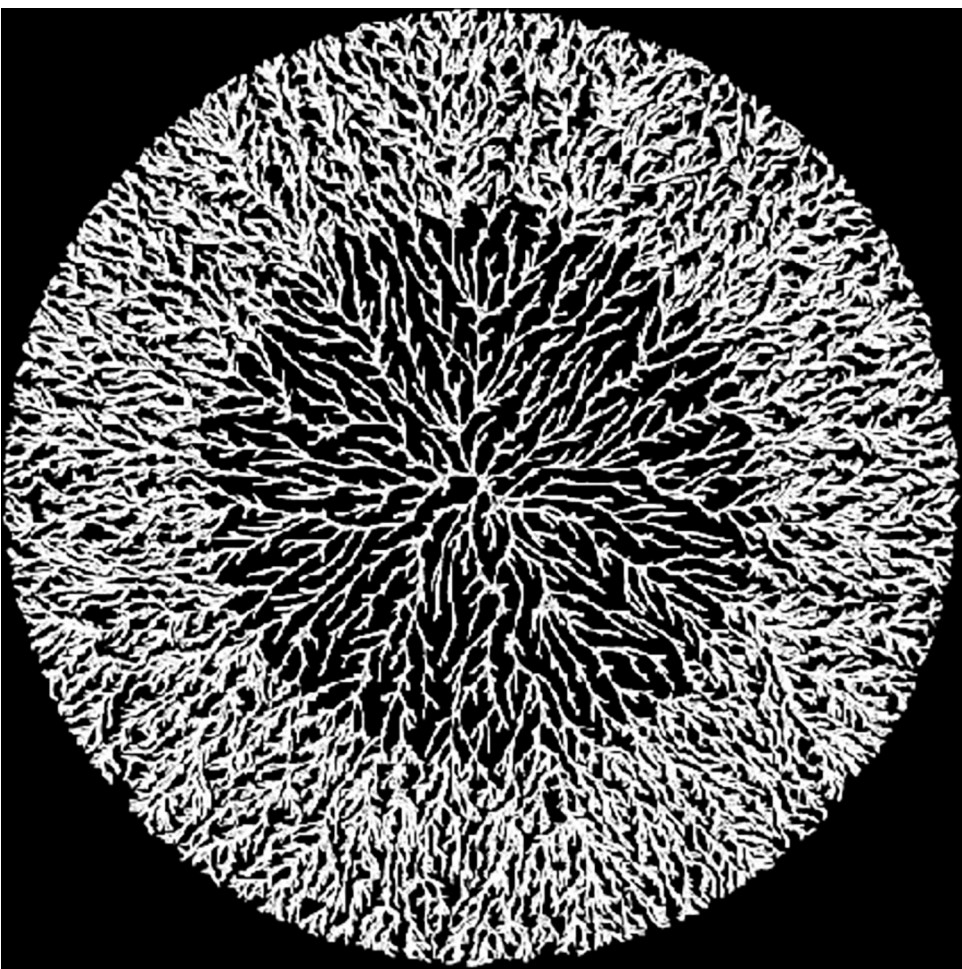

**Fig 12. Annular growth patterns caused by a change in mobility.** The radius of the disk is 11.3 units. Particles within 6.5 units of the center have mobility of 200. Those further away have mobility of 20. This simulates a change in drying time during polymer pattern formation. Longer drying times cause more extensive polymer aggregation and contraction resulting in a lower fractal dimension. Remaining model parameters are: *bond-range* = 0.25, and *bond-rate* = 0.1.

more than 6.5 units from the center. Thus, the center models a longer drying time meaning more ability for the polymer to contract. The similarity between our model and the polymer system suggests that the mobility parameter simulates the experimental drying time.

The other phenomenon our model replicates with regard to polymer morphology is the rearrangement of polymer thin films when they are exposed to water and then allowed to reform, a process known as solvent annealing. The first two panels in Fig 13 show a polymer before and after solvent annealing. The third panel is an overlay of the two images for comparison. The applied water appears to have liberated a portion of the particles, modifying the original structure. When the water is removed, the liberated polymer reforms. The new growth is less curvy than the original structure suggesting an increase in *bond-range* as shown in Fig 4. The soluble portion of the polymer is less crystalline and differs from the insoluble portion. This difference most likely gives rise to different *bond-range* values between the original and the new structures.

We model annealing in three stages. First, the normal model was run with the parameters set to replicate a polymer with the dimension and structure seen in the first panel of Fig 13



**Fig 13. Spin-coated fractals before and after solvent annealing.** Optical images (500x) of dewetted patterns obtained from adsorptive spin-coating of 400 μL PVOH[99%H] solution on PDMS[49k] substrates (1.4 cm x 1.4 cm) at 900 rpm before and after solvent annealing. Solvent annealing involved soaking samples in Milli-Q water for 1 h followed by drying under nitrogen gas. Overlaid optical image depicts polymer rearrangement as the result of solvent annealing—before image in magenta and after image in green. Fractal dimensions were 1.58 (+/- 0.07) before solvent annealing and 1.64 (+/- 0.07) after solvent annealing.

(before annealing). Second, we removed all of the bonds as well as a fixed proportion of randomly selected particles from the structure and allowed the original structure to reform. The removed particles were randomly placed on the original disk. This simulates the liberating of some of the polymer into solution while some of the original structure is retained. Finally, all external links were set to active, and the structure was allowed to finish growing. Two growth stages were necessary after annealing as it is hypothesized that many of the original bonds are maintained during annealing and many of the particles have limited mobility. Fig 14 shows the results of such a simulation. A similar deformation of the original polymer combined with higher-dimensional new growth is apparent. Fractal dimensions for the polymer before and after annealing were a bit higher than the average values observed experimentally, but still capture an increase in fractal dimension after annealing (see Fig 13).

## Discussion

### Similarities and differences with other models of fractal growth

There are many mathematical growth models that exhibit a fractal dimension and produce structures that resemble empirical phenomena. Some are quite simple, such as the Eden model [15] and the DLA model [19], others are of medium complexity such as the Born model of fracture growth [17], the dielectrical breakdown (DB) model [8], and the model of bacterial growth of Ben-Jacob et al. [6], while others are numerically if not conceptually quite complex such as the mathematical model for Hele-Shaw cells [35,36] or phase models of ice development (e.g. 25). We feel that our model is comparable to the DLA model in presenting a

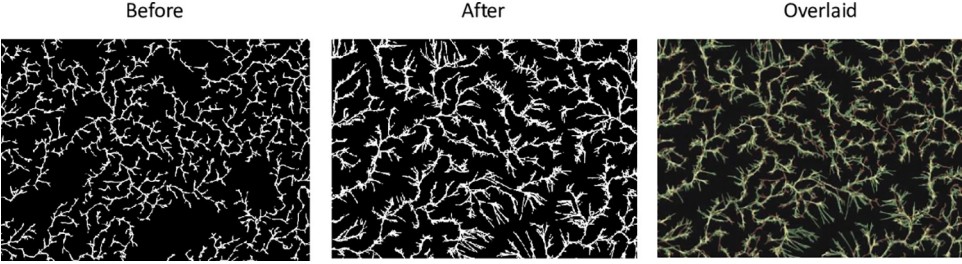

**Fig 14. Model output before and after simulated solvent annealing.** Parameters for the left model were *bond-range* = 0.2, *bond-rate* = 0.02. and *mobility* = 100. 30% of the particles were liberated, and after annealing *bond-range* was increased to 0.5. The disks were cropped to replicate the images in Fig 13. Fractal dimensions were 1.71 and 1.75, slightly higher than those in Fig 13. Coloration of the before and after components in the overlaid image is the same as in Fig 13.

conceptually straightforward model of growth. Computationally, the contraction step adds in some complexity but still remains significantly simpler than the numerical approximations required for the Born model or the DB model.

The variability in output of our model is comparable to other models. The bacterial growth model of Ben-Jacob et al. [6] also has two parameters that dictate model output, producing a range of behaviors not dissimilar from ours (see Fig 3 in [6]). And variations to the DLA model in terms of the probability distribution of incoming particles and random walk step size also produces a range of behaviors (see Fig 5 in [20]). Variability in these models is comparable to ours in that much of the variation concerns the number of branches and how dendritic the growth is in appearance. However, neither model is able to produce the kind of random walk behavior seen in our model output with low dimension.

One significant difference between our model and most fractal growth models is that mathematically, our structure is a triangular network with vertices of zero mass. In other words, our model produces bonds between particles rather than modeling the aggregation of particles themselves. This is not entirely unique. The Born model output consists of the bonds that break as the fracture spreads. L-system models produce geometric structures by converting a sequence of strings into a series of commands for drawing straight line segments, in effect producing a network [37,38]. In terms of polymer pattern development, the target of simulation for our model is the same as if the DLA model were being utilized, but the structure is being modeled as an expanding mass of bonds rather than an aggregating mass of particles.

Our triangular bond model simulates multi-body interactions in forming a realistic fractal network. A new particle in the vicinity of a growing front experiences multiple, attractive, intermolecular forces and can get pulled in to bond to the structure. Bond-range is adjustable to scale with the nature of intermolecular forces. Formation of solid structures, such as those prepared by adsorptive spin-coating of polymer solution, often involves solvent evaporation. Mobility simulates variation in drying time and extent of structure contraction. The key features in our model–multi-body bonding, bond-range, and mobility–are drawn upon physical parameters in real applications.

Finally, we would note that while the model we present here is inherently a 2D, planar model, it is readily extendable to higher dimensions. In three dimensions, bonds are formed by connecting a new particle to each vertex of an external triangular face on the growing mass. Contraction occurs precisely as with the 2D model, and initial runs we have conducted also show fractal growth. Extensions to higher dimensions would be analogous.

## Models with variable dimension

The variability of the fractal dimension of our growth model is an important feature, especially since the dimension can be tuned with the model parameters. While the original DLA model appears to have a fractal dimension that is consistently 1.71 [19,21], variations of the model do exhibit other dimensions, and some exhibit dimensions that vary with the parameters. The generalized DLA model with different dimensional random walks achieves dimensions between 1.71 and 1.95 [22], and allowing the aggregates to also diffuse (cluster aggregation) reduces the dimension to approximately 1.5 [4,23]. DB models, which can be considered to be an extension of the DLA model, exhibit a variable and tunable fractal dimension between 1.6 and 2 dependent upon the strength of the interaction with the electrical field [8].

Variable dimension often comes at the cost of increased model complexity, both conceptually and computationally. Ehrl, Soos, and Lattuada [39] were able to produce clusters in 3D of any desired dimension from 2.2 to 3 by using an aggregation algorithm specifically focused on producing the desired dimension. The algorithm has value for modeling observed

aggregations, however it is significantly more complex than aggregation models such as DLA and the DB model, and it lacks a direct connection to a known aggregation process [40]. Similarly, the bacterial growth model of Ben-Jacob et al. [6] successfully replicates bacterial growth patterns and has variable dimension less than 1.7. But again, model dynamics are quite complex. Perhaps the simplest model with tunable dimension is the DB model, but this model requires an iterative solution to the Laplace equation after each particle docks.

By comparison, our model is conceptually simple and also has a tunable dimension although with significant variability between model runs at low and medium dimensions. We are working on algorithms that would enable runs with a greater number of particles to determine the degree to which the stochasticity in dimension is inherent in the model. Having the ability to adjust dimension increases our ability to fit the model to specific phenomena as well as to test hypotheses related to those phenomena. This is demonstrated above by our ability to adjust dimension to match polymers with different drying times and consequently different fractal dimensions. Further, by associating such changes with specific model parameters, we can craft hypotheses about what elements of the system cause the change in dimension.

## Comparable applications to polymers and other phenomena

Fractal growth models like ours have been used to model a wide variety of phenomena [2]. Zhang and Liu [12] modeled viscous fingering in liquids with varying fractal dimension using a modified DLA model. Other modifications of the DLA model have been used to model chemical vapor deposition [41], urban migration and growth patterns [11], general dendritic growth [13], and the evolution of river networks [9]. L-systems have been used to model many biological systems [37] including the growth of mycorrhizal fungi [38].

Polymer pattern development in particular has been modeled by several different fractal growth models. Gao et al. [3] demonstrated a similarity between fullerene doped polymer thin films and the cluster diffusion model [23]. Similarly, Amir, Ali, and Mohamed [42] modeled fractals in polymer electrolyte films using the DLA model. Morozova et al. [43] used a clustering model for charged nanoparticles that has good agreement with observed colloidal clusters and networks. On a finer scale, Monte Carlo based methods have been used to model individual polymer chains as lattice animals, a set of connected vertices on a hypercubic lattice. [44,45].

In a similar vein, we have demonstrated the ability of our model to create structures very similar to those produced by the spin coating of polymer thin films and subsequent dewetting. We are not only able to match the dimension and general shape of these polymers, but also, by making reasonable changes to model parameters, we are able to account for differences in polymer structure caused by variations in drying time or the application of a solvent and the resultant annealing. This suggests a deeper connection between our model and the dynamics of polymer crystallization and rearrangement, something we hope to explore in future work. Ideally, our model will be able to make predictions for polymer features under different conditions as well as lead to new hypotheses regarding changes to polymer dynamics under different experimental treatments.

More importantly, given the fundamental differences in approach between our model and other fractal growth models, it is our hope that this model will allow for the relatively simple and efficient simulation of experimental systems that are not amenable to being modeled with simple, aggregation-based models. We would add that our model can easily be extended to three dimensions where the activated "links" are triangles on the surface of the growing structure which connect to unattached particles by forming bonds between the particle and all three vertices of the triangle forming a tetrahedron. The structure of these 3D fractals is the subject of ongoing work.

## Conclusion

The power of the diffusion limited aggregation model of Witten and Sander [18] lies in its tractability and simplicity. It is a model that can be effectively described to a wide audience and easily implemented in many different platforms. And yet, its behavior is varied and complex and able to simulate a wide variety of empirical phenomena. In this work, we have introduced a model of comparable (albeit greater) complexity that takes a fundamentally different approach to fractal growth, what might be dubbed a "reach-and-grab" model as opposed to a "sit-and-wait" model. Our model also exhibits a range of complex behaviors that appear similar to a variety of real-world phenomena. Model output demonstrates variability in fractal dimension that is correlated with model parameters. This leads us to hope that for specific situations like the development of polymer fractals using spin-coating, the model can be tuned to simulate specific features of the system and ideally lead to predictive models for yet-to-be-produced fractal structures.

## Supporting information

**S1 File.**
(XLSX)

**S2 File.**
(XLSX)

**S3 File.**
(XLSX)

## Acknowledgments

We wish to thank two reviewers whose comments and critiques improved the work presented here.

## Author Contributions

**Conceptualization:** Kenneth Mulder, Wei Chen.

**Formal analysis:** Kenneth Mulder, Wei Chen.

**Funding acquisition:** Wei Chen.

**Investigation:** Sophia M. Lee.

**Methodology:** Kenneth Mulder, Sophia M. Lee.

**Project administration:** Kenneth Mulder, Wei Chen.

**Supervision:** Wei Chen.

**Validation:** Kenneth Mulder, Wei Chen.

**Visualization:** Kenneth Mulder, Sophia M. Lee.

**Writing – original draft:** Kenneth Mulder.

**Writing – review & editing:** Wei Chen.

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
