## [Decision Letter · Decision Letter 0]

17 Nov 2023

PONE-D-23-28024A triangular model of fractal growth with application to adsorptive spin-coating of polymersPLOS ONE

Dear Dr. Mulder,

Thank you for submitting your manuscript to PLOS ONE. After careful consideration, we feel that it has merit but does not fully meet PLOS ONE’s publication criteria as it currently stands. Therefore, we invite you to submit a revised version of the manuscript that addresses the points raised during the review process. As you can see below, both reviews are enthusiastic, although some improvements are required. Please read the Reviewers' comments carefully and respond to all questions and comments in the revised version of the manuscript.

We look forward to receiving your revised manuscript.

Kind regards,

Malgorzata J. Krawczyk

Academic Editor

PLOS ONE

“The adsorptive spin-coating work is financially supported by the National Science Foundation (DMR-1807186).”

“SL was funded by the National Science Foundation, grant number DMR-1807186 (nsf.gov).  The funders had no role in study design, data collection and analysis, decision to publish, or preparation of the manuscript.”

4. We note that Figures 9 and 11 in your submission contain copyrighted images. All PLOS content is published under the Creative Commons Attribution License (CC BY 4.0), which means that the manuscript, images, and Supporting Information files will be freely available online, and any third party is permitted to access, download, copy, distribute, and use these materials in any way, even commercially, with proper attribution. For more information, see our copyright guidelines: http://journals.plos.org/plosone/s/licenses-and-copyright.

a. You may seek permission from the original copyright holder of Figures 9 and 11 to publish the content specifically under the CC BY 4.0 license.

Reviewers' comments:

Reviewer's Responses to Questions

**Comments to the Author**

1. Is the manuscript technically sound, and do the data support the conclusions?

Reviewer #1: Yes

Reviewer #2: Yes

2. Has the statistical analysis been performed appropriately and rigorously? 

Reviewer #1: No

Reviewer #2: Yes

3. Have the authors made all data underlying the findings in their manuscript fully available?

Reviewer #1: Yes

Reviewer #2: Yes

4. Is the manuscript presented in an intelligible fashion and written in standard English?

Reviewer #1: Yes

Reviewer #2: Yes

5. Review Comments to the Author

Reviewer #1: This work presents a stochastic aggregation model that produces dendritic structures DLA-like following a simple set of rules. The model is, in principle, interesting and seems to replicate the morphology of certain polymeric structures. There is real value in novel models like the one presented, as they provide researchers with tools to try different growing scenarios, so I would like to see this work published. However, some major issues need to be clarified/corrected before publication.

1. In the introduction, the wording is confusing: What are the normal rules of geometry? Do the authors mean Euclidean Geometry (line 44)? Which objects do they refer to generally having dimensions 1,2, or 3 (lines 44-45)? Objects in the physical world? Hausdorff dimension can be generalised to Rn, so it is important to clarify this point.

2. Which type of fractal dimension is discussed here? Capacity? Correlation? Information? I guess it’s capacity (box-counting), but please confirm.

3. The claim that this model is comparable in simplicity to the DLA does not hold. In its original definition, the DLA starts with a seed, and a particle undergoes a random walk until it “touches” the seed. The process repeats with one particle walking at each iteration. That’s it, no more parameters. The presented model contains several parameters, which create a non-trivial phase space to analyse. There are some suggestions in this work of what to expect if some parameters change (section III.A) and Figure 5, but nothing systematic. The authors must study the phase space to understand this model and its full potential. I am not talking about the full combinations of all the elements of Table 1, but at least Probability of bonding and Maximum bonding distance, in my view, are the critical parameters. Mobility period also seems quite important, but it is unclear from the text how it works (more about this later). Authors need to perform several runs for many different combinations of these parameters. For example, run 100 experiments with bond-rate = i and bond-range = j, where i in [0.01,0.02,…,0.1] and j in [0.1,0.2,0.3,…,1], i.e., 100 for bond-rate=0.01 and bond-range=0.1; 100 experiments for bond-rate=0.01 and bond-range=0.2, and so on. At the end, you’ll have 10,000 experiments to study this limited phase space. Only then can we start to understand the model. Each experiment would produce the four statistics from section III.B statistics, and then the authors need to get the mean value for each combination.

4. The model definition is acceptable, except for the contraction step. I understand the general idea behind the parameter, but the mobility concept is unclear. How are the linked neighbours defined? This line, “a displacement equal to 0.01 times each of the vectors represented by directed links from the particle to each of their linked neighbors,” makes little sense. What is 0.01 time in this context? Maybe it is only how it is redacted, but it needs to be explained better.

5. In the introduction, there is a strong emphasis on the fractal nature of these models. However, this characteristic dilutes as the work advances, only to be mentioned again in section III.B, and there is no mention of why it’s relevant for this type of aggregates.

6. Besides the lack of a rigorous statistical treatment of the model, the other major problem of this piece is how the fractal dimension is calculated. Working over the images creates a serious bias in the values. The reported dimension has a strong correlation with the image resolution. As the authors generate the model, they have all the exact positions for each particle, so they need to use the box-counting using these positions. There are plenty of codes for doing that; no need to create an image and pass it through ImageJ. In a revised version, I suggest sharing such code or a pseudo-code as part of the manuscript.

7. The text mentions the concept of Lacunarity, which is never defined or cited. I fail to see the value of the Lacunarity measures in this work in its present form. I suggest removing it completely or properly defining and better explaining its importance for the model.

8. Better images are needed. Figure 1 is good, except, again, at the contraction frames. I suggest showing a zoom section for Figures 2 and 3 to visualise the actual triangles and the local structures they form.

9. At line 500, it is mentioned that variable dimension often comes at the cost of increased complexity. What does this mean? That Higher dimension � Higher complexity? Or vice versa. How is complexity defined? These are not trivial questions. I suggest removing all lines from 500 to 509, as those add nothing but confusion to this work.

Reviewer #2: The manuscript introduced a new, interesting algorithm to create fractal patterns. The algorithm will start by grouping particles in triangular patterns, with each side of a triangle possibly capturing another particle. Then, the patterns are contracted. The algorithm's flexibility relies on changing the probability and distance of particles being caught and the mobility of the patterns, i.e., the contraction rate of the patterns compared to the capture of new particles. The authors have shown how various fractal patterns with different fractal dimensions can be obtained. Comparison with spin coating of polymer shows qualitative agreement of the patterns.

The work is very interesting and novel. The types of pattern formed by the algorithm looks very useful for describing complex phenomena, such as spin coating and drying patterns. The description is clear, and the examples are convincing. I want the authors to address a couple of minor issues before publication.

1) It would be interesting to have a better look at the early stages of the pattern formation, shown in Figures 2 and 3. What are the authors representing in Figures 2 and 3? The particles or the bonds?

2) Would it be possible to compute quantities like a radial distribution function, which could provide additional information about the fractal structure of the patterns?

3) Did the authors try to extend the work to 3 dimensional systems, using tetrahedra instead of triangles?

6. PLOS authors have the option to publish the peer review history of their article (what does this mean?). If published, this will include your full peer review and any attached files.

Reviewer #1: **Yes: **Roberto Murcio

Reviewer #2: No

---

## [Author Response · Author response to Decision Letter 0]

21 Dec 2023

Specific responses to reviewer comments are detailed in our rebuttal letter.

---

## [Decision Letter · Decision Letter 1]

1 Feb 2024

A triangular model of fractal growth with application to adsorptive spin-coating of polymers

PONE-D-23-28024R1

Dear Dr. Mulder,

We’re pleased to inform you that your manuscript has been judged scientifically suitable for publication and will be formally accepted for publication once it meets all outstanding technical requirements.

Kind regards,

Malgorzata J. Krawczyk

Academic Editor

PLOS ONE

Additional Editor Comments (optional):

Reviewers' comments:

Reviewer's Responses to Questions

**Comments to the Author**

1. If the authors have adequately addressed your comments raised in a previous round of review and you feel that this manuscript is now acceptable for publication, you may indicate that here to bypass the “Comments to the Author” section, enter your conflict of interest statement in the “Confidential to Editor” section, and submit your "Accept" recommendation.

Reviewer #1: All comments have been addressed

Reviewer #2: All comments have been addressed

2. Is the manuscript technically sound, and do the data support the conclusions?

Reviewer #1: Yes

Reviewer #2: Yes

3. Has the statistical analysis been performed appropriately and rigorously? 

Reviewer #1: Yes

Reviewer #2: Yes

4. Have the authors made all data underlying the findings in their manuscript fully available?

Reviewer #1: No

Reviewer #2: Yes

5. Is the manuscript presented in an intelligible fashion and written in standard English?

Reviewer #1: (No Response)

Reviewer #2: Yes

6. Review Comments to the Author

Reviewer #1: (No Response)

Reviewer #2: (No Response)

7. PLOS authors have the option to publish the peer review history of their article (what does this mean?). If published, this will include your full peer review and any attached files.

Reviewer #1: **Yes: **Roberto Murcio

Reviewer #2: No

---

## [Editor Report · Acceptance letter]

13 Feb 2024

PONE-D-23-28024R1 

PLOS ONE

Dear Dr. Mulder, 

I'm pleased to inform you that your manuscript has been deemed suitable for publication in PLOS ONE. Congratulations! Your manuscript is now being handed over to our production team.

Kind regards, 

on behalf of

Dr. Malgorzata J. Krawczyk 

Academic Editor

PLOS ONE